# A cross-sectional study on the impact of early systematic antidepressant therapy on positive/negative affect and impulsivity in euthymic bipolar disorder patients

Dongyu Han[1,2‡], Peng Cui[1‡], Mingjin Wang[2], Xiaofei Hou[1], Chenghao Yang[1]*, Jie Li[1,2]*

1 Tianjin Mental Health Center, Tianjin Anding Hospital, Tianjin, China, 2 Tianjin Medical University, Tianjin, China

‡ The two authors share the first authorship on this work.
* yts83420@163.com (CY); jieli@tjmhc.com (JL)

## Abstract

### Background and objectives

Due to an initial misdiagnosis, bipolar patients who experienced a depressive episode as their first onset were often treated with antidepressants, and continued to exhibit sleep disturbances and elevated impulsivity, even during periods of euthymia. The study aims to assess the effect of systematic antidepressant treatments in the early stages on impulsivity and positive/negative affect in patients with bipolar euthymic disorder. Additionally, it explores the potential mediating effects of positive/negative affect on the relationship between antidepressant uses and impulsivity.

### Methods

A cross-sectional study was conducted involving 124 Han Chinese patients with bipolar disorder (BD), who were equally divided into two groups based on a history of systematic antidepressant treatments in the early stages: the systematic antidepressant treatment (AT) group and the no antidepressant treatment (NT) group. Participants were assessed using the Positive and Negative Affect Scale and the Barratt Impulsiveness Scale. Statistical analyses included Chi-square tests, t-tests, Mann-Whitney U tests, and mediation analysis using bootstrapping.

### Results

Patients in the AT group exhibited significantly higher levels of negative affect (p = 0.017), attentional impulsivity (p = 0.035), non-planning impulsivity (p = 0.010), and total impulsivity (p = 0.011) compared to those in the NT group. No significant differences were found in positive affect or motor impulsivity between the two groups. There was a significant inverse correlation between negative affect and

**Data availability statement:** All relevant data are within the manuscript and its Supporting information files.

**Funding:** The study was financially supported by funds from "Research Plan Project of Tianjin Municipal Education Commission" (2022KJ264), "Tianjin Key Medical Discipline Construction Project" (TJYXZDXK-3-015B), and "Beijing-Tianjin-Hebei Basic Research Cooperation Project" (23JCZXJC00230, J230011).

**Competing interests:** The authors have declared that no competing interests exist.

motor impulsivity ($p < 0.01$). Mediation analysis indicated that negative affect did not play a significant mediating role between systematic antidepressant treatments and impulsivity.

## Conclusions

Systematic antidepressant treatment in the early stages is associated with increased negative affect and impulsivity during bipolar euthymia. These findings highlight the importance of cautious antidepressant prescription and the need for early diagnosis and personalized treatment strategies for BD patients. Longitudinal research is warranted to further elucidate the relationships between antidepressant use, affect state and impulsivity.

---

## Introduction

Bipolar disorder (BD) is a significant, enduring mental health condition marked by intertwining depressive, manic, and hypomanic episodes, with a high global lifetime prevalence rate exceeding 1% [1]. A frequent comorbidity with BD is the high impulsivity, which can lead to negative outcomes, including suicidality. Indeed, the suicide rates in individuals with BD are reported to be 20–30 times higher than those in the general population [2,3]. Studies have shown that impulsivity scores, both overall and across specific dimensions of the Barratt Impulsiveness Scale (BIS-11A), are typically higher in BD patients compared to the general population [4,5]. even after achieving euthymia [6]. Moreover, motor impulsivity is more pronounced in BD patients than in those with major depressive disorder (MDD) [4]. High impulsivity may partly account for the increased risk of suicidality in individuals with BD. Individuals with BD often struggle with emotional dysregulation and emotion-associated activity [7,8], which could amplify impulsivity and heighten the likelihood of engaging in high-risk behaviors such as self-harm and suicide [9], BD is characterized by dramatic shifts between positive and negative emotions or by a mixed state, which can impact decision-making and stimulate impulsive responses. Emerging research indicates that impulsive reactions to emotions are stronger predictors of suicidality than other forms of impulsivity [2]. For instance, BD patients appear especially prone to impulsive behavior when experiencing intense positive emotions [10]. which correlates with suicide ideation, attempts, and self-harm behaviors [2]. In contrast, other studies have shown that greater negative affect, but not positive affect, predicts subsequent increases in impulsivity [7,11]. It seems that both negative and positive affect contribute to increased suicidality in BD patients by stirring impulsivity. However, inconsistent to prior research, a meta-analysis found that BIS-11A subscales did not significantly relate to self-harm or suicidal behaviors in BD patients [12]. Considering these inconsistencies, more studies are needed to explore the relationship between impulsiveness and positive/negative affect in BD individuals to promote a more in-depth understanding of suicidal ideation and attempts, as well as the potential risk factors that remain undefined.

Many BD patients do not exhibit manic or hypomanic symptoms in the early stages (defined as a period from the initial depressive onset to the diagnosis of BD), contributing to a delayed diagnosis by 5–10 years [13], which may lead to inadequate interventions with antidepressants. Empirical evidence indicates that antidepressant treatment, particularly when used as monotherapy, can destabilize the natural course of BD, precipitating episodes of mania, hypomania, or mixed states, ultimately manifesting as emotional instability and enhanced impulsivity. A network meta-analysis indicated that while antidepressants are generally effective for bipolar depression, they carry a higher risk of manic switching compared to antipsychotics [14]. One cross-sectional study showed that, out of 123 euthymic BD patients, 58 experience at least one manic switching under antidepressant treatment [15]. Expanding beyond patients with bipolar depression, subthreshold mood switches frequently occur in unipolar depression during acute antidepressant treatment and the continuation phase, which is associated with more severe suicidal behavior ($p < 0.001$) [16].

The issues of emotional instability and impulsivity, even during the period of euthymia, may increase the occurrence of risky behaviors in individuals with BD, thereby adding challenges and burdens to clinical treatment and disease management. Identifying and addressing these issues can contribute to the development of more effective intervention strategies for this condition. The present study aimed to assess the influence of systematic antidepressant treatments (receiving an adequate dosage of antidepressants for at least 6 weeks) in the early stages on impulsivity and positive/negative affect in euthymic BD patients with stringent criteria. For instance, BD patient must initiate with a depressive episode, and the age range was narrowed to between 18 and 50 years old. Furthermore, the study also explored the potential mediating effect of positive/negative affect on the relationship between antidepressant use and impulsivity in this group of individuals.

## Materials and methods

### Setting and participants

This is a cross-section study conducted in the outpatient and inpatient departments of Tianjin Anding Hospital located in Tianjin, China, from 1st September, 2020–30th May, 2023. This study was conducted in accordance with the ethical guidelines of Declaration of Helsinki. The study had been approved by the Medical Ethics Committee of Tianjin Anding Hospital (approval No. LKSKD 2020-18; approval date, 25th Aug, 2020). Informed consent was signed by all participants when they were enrolled. The study recruited two groups of participants: the "systematic antidepressant treatment" (AT) group and the "no systematic antidepressant treatment" (NT) group (n = 62 respectively). Participants were assigned based on whether they had received systematic antidepressant treatments during the early stage of their condition. The definition of "systematic antidepressant treatment" is that patients receive treatment with Selective Serotonin Reuptake Inhibitors (SSRIs) or Serotonin-Norepinephrine Reuptake Inhibitors (SNRIs) as monotherapy, including sertraline 50–200 mg/d, fluoxetine 20–60 mg/d, paroxetine 20–60 mg/d, citalopram 20–60 mg/d, escitalopram 10–20 mg/d, venlafaxine 75–225 mg/d, and duloxetine 30–120 mg/d, for at least 6 weeks. No restrictions were placed on benzodiazepines, zolpidem, zopiclone, or zaleplon, used as adjuncts to improve sleep. The screening process was conducted by two senior psychiatrists, and the following rating works were handed over to other investigators, and the subsequent ratings were performed by other investigators, who were blinded to the participants' group assignments in order to minimize bias in evaluation and analysis. The inclusion criteria were as follows: a diagnosis of BD according to the Diagnostic and Statistical Manual of Mental disorders, Fourth Edition (DSM-IV-TR) diagnosed with Structured Clinical Interview for DSM-IV (SCID) (the study was conceived in 2019 and launched in mid-2020, at which point the DSM-5 and SCID-5 had not yet been implemented in China); a current state of euthymia, defined scores on both the Montgomery-Åsberg Depression Scale (MADRS) and Young Manic Rating Scale (YMRS) being less than 7 [17], and lasting for at least 4 weeks before recruitment; initiated with a depressive episode confirmed by the MINI-International Neuropsychiatric Interview (MINI); aged between 18–50 years; compliance with all study procedures; and the ability to sign the informed consent. The exclusion criteria were as follows: a history of psychotic symptoms; any other mental disorders according to the DSM-IV-TR; history of substance dependence/abuse; severe physical disease that impose a significant psychological and/or economic burden on

participants, such as kidney and liver failure, uncontrolled hypertension, cardiovascular, cerebrovascular and pulmonary disease, thyroid disease, diabetes. All participants were screened with Hypomania Check List-32 items (HCL-32) to detect the potential hypomanic symptoms in the early stages. Additionally, histories of diagnosis and treatments were confirmed using medical records and information from their guardians. All participants completed the "Positive and Negative Affect Scale" (PANAS) and BIS-11A, and general demographic information was collected, including age, sex, ethnicity, age of onset, years of delayed diagnosis, and use of mood stabilizers. The entire procedure was finished within 180 minutes (see Fig 1. Study flowchart).

## Measurements of affective symptoms and impulsivity

The positive and negative affect was assessed using the PANAS [18]. The euthymic state was evaluated using the YMRS and MADRS, with both scores being less than 7 [11]. The BIS-11A was used to assess impulsivity, including attentional, non-planning, and motor impulsivity [4].

## Statistical analysis

SPSS (version 22.0) was adopted for data analysis. Means and standard deviations (SD) were used to describe continuous variables, and numbers (n) and percentages (%) were used for categorical variables. Differences between the two groups in demographic and clinical variables were compared using the chi-square ($\chi^2$), t-test, or Mann-Whitney U tests, depending on the nature and distribution of variables. To examine mediating effects, the bootstrapping technique using Model 4 of the PROCESS function was employed, with 5000 bootstrap estimates and bias-corrected 95% confidence intervals (CI) generated. Correlation analysis between any two variables of antidepressant use, affective state and

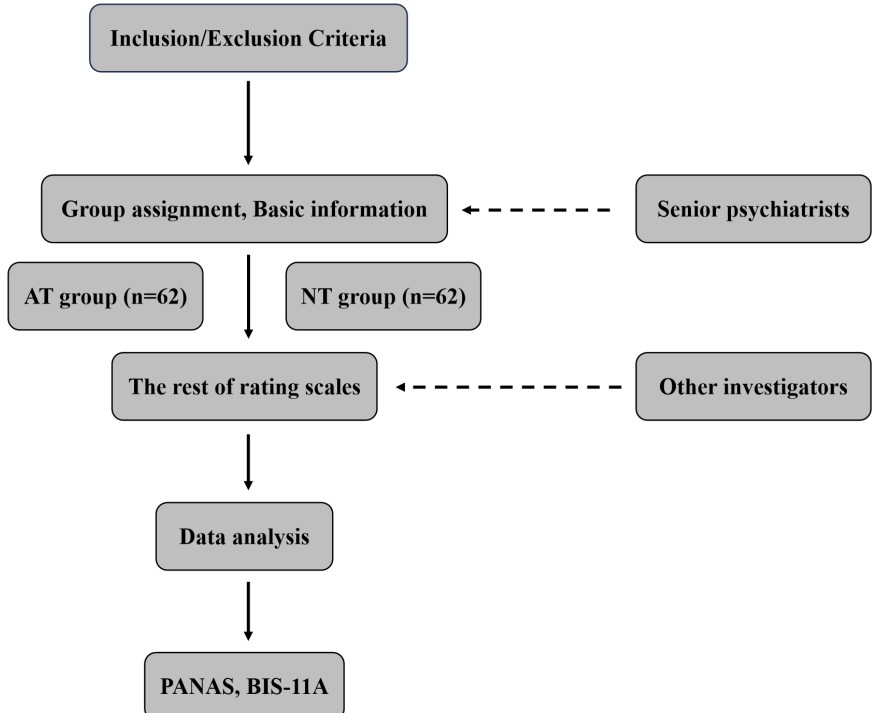

**Fig 1. Study flowchart.** AT, systematic antidepressant treatment; NT, no systematic antidepressant treatment; PANAS, Positive and Negative Affect Scale; BIS-11A, Barratt Impulsiveness Scale 11-A.

impulsivity using the Spearman correlation test. To reduce the effect of confounding factors, we adjusted for the covariates such as sex, age, years of delayed diagnosis, and use of mood stabilizers in the mediation analysis. All statistical tests were performed using two-tailed comparisons, with statistical significance set at p < 0.05.

## Results

### Participant demographics and characteristics

A total of 124 Han Chinese patients with BD were included and divided into the AT group and NT group (n = 62 for each group). The scores of the YMRS and MADRS did not show significant differences between the two groups (both p > 0.05). The distributions of age and sex in the two groups were not significantly different (AT vs NT, age: 29.8 ± 9.1 vs 30.9 ± 9.1, p = 0.476; sex: $\chi^2$ = 0.300, p = 0.584). There were also no significant differences between the groups regarding years of delayed diagnosis (AT vs NT, 4.5 ± 4.2 vs 5.1 ± 6.2, Z = −0.314, p = 0.753) or use of mood stabilizers ($\chi^2$ = 0.238, p = 0.625). The details are shown in Table 1.

### The comparison of positive/negative affect between AT and NT groups

The effect of antidepressant use on positive and negative affect was investigated using PANAS. The AT group had a higher level of negative affect than the NT group (AT vs NT 24.9 ± 6.6 vs 22.1 ± 6.1, p = 0.017), which was statistically significant, even after correcting for sex, age, years of delayed diagnosis, and use of mood stabilizers. There was no significant difference between the groups in positive affect (AT vs NT 16.4 ± 7.6 vs 15.9 ± 5.5, p = 0.684). See Table 2 for details.

### The level of impulsivity in AT and NT groups

The features of impulsivity in the two groups were measured by BIS-11A. Attentional impulsivity was stronger in the AT group than in the NT group, and the difference was statistically significant (AT vs NT: 13.3 ± 2.9 vs 12.1 ± 2.8, Z = −2.106, p = 0.035). There was also a significant difference in non-planning impulsivity between the two groups (AT vs NT: 25.0 ± 4.9 vs 22.7 ± 4.8, t = −2.603, p = 0.010), as well as in the total score of BIS-11A (AT vs NT: 61.4 ± 9.4 vs

**Table 1. Sociodemographic and clinical characteristics.**

| Variables | NT (n = 62) | AT (n = 62) | t/Z/$\chi^2$ | p |
|---|---|---|---|---|
| Age | 30.00 (23.75-37.00) | 28.50 (22.00-35.25) | −0.713 | 0.476 |
| Sex (women /men) | 38/24 | 35/27 | 0.300 | 0.584 |
| Diagnostic delay (years) | 3.00 (1.00-6.00) | 3.00 (1.00-7.00) | −0.314 | 0.753 |
| Mood stabilizer(yes/no) | 9/53 | 11/51 | 0.238 | 0.625 |
| Depressive symptoms (MADRS) | 4.00 (1.00-7.00) | 3.00 (0.00-6.00) | −0.930 | 0.352 |
| Manic symptoms (YMRS) | 4.00 (3.00-5.00) | 4.00 (2.00-5.00) | −0.931 | 0.352 |

Note. MADRS, Montgomery-Asberg Depression Rating Scale; YMRS, Young Mania Rating Scale; AT, systematic antidepressant treatment; NT, no systematic antidepressant treatment

**Table 2. The comparison of positive/negative affect between AT and NT groups.**

| Variables | NT (n = 62) | AT (n = 62) | t/Z/$\chi^2$ | p |
|---|---|---|---|---|
| Positive affect | 14.00(12.00-19.00) | 14.00(11.00-18.00) | −0.407 | 0.684 |
| Negative affect | 22.13 ± 6.142 | 24.90 ± 6.558 | −2.431 | 0.017 |

Note: AT, systematic antidepressant treatment; NT, no systematic antidepressant treatment.

57.2 ± 8.6, t = −2.588, p = 0.011). However, the level of motor impulsivity did not show a statistically significant difference between the groups (AT vs NT: 23.1 ± 4.0 vs 22.4 ± 3.8, Z = −1.417, p = 0.156). The statistical significances above remained after adjusting for sex, age, years of delayed diagnosis, and use of mood stabilizers. The detailed information is presented in Table 3.

## Analysis on the mediation effects of positive/negative affect

The study investigated whether positive or negative affect mediated the relationship between systematic antidepressant treatments and impulsivity in bipolar euthymic patients. We first conducted a Spearman correlation analysis, which indicated a clear positive link between the use of antidepressants and increased negative affect (r = 0.216, p = 0.016), although no such association was found between antidepressant use and positive affect (r = −0.037, p = 0.686). Likewise, significant associations were observed between antidepressant use and various forms of impulsivity, including attentional impulsivity (r = 0.190, p = 0.035), non-planning impulsivity (r = 0.228, p = 0.011), and overall impulsivity (r = 0.245, p = 0.006), as per the BIS-11A scale, except for motor impulsivity (r = 0.128, p = 0.157). Additionally, significant correlations were noted between positive affect and the overall impulsivity score (r = −0.114, p = 0.209), and negative affect and motor impulsivity (r = −0.260, p = 0.004), but no such associations were found between other variables (see Table 4 for details). However, we failed to construct a complete mediating relationship from antidepressant use to impulsivity through affective states (all p > 0.05), although some of the direct, indirect, and total effects were statistically significant, such as the total effect of antidepressant use on overall impulsivity (Effect$_{total}$ = 4.357, p = 0.008), and the direct effect of antidepressant uses on full impulsivity (Effect$_{direct}$ = 4.497, p = 0.005) (see Table 5 for details of mediation analyses). Combining the analyses of mediation effect and Spearman correlation tests, the results suggest that an increase in negative affect does not necessarily lead to higher impulsivity in bipolar euthymic patients.

**Table 3. The level of impulsivity in AT and NT groups.**

| Variables | NT (n = 62) | AT (n = 62) | t/Z/χ² | p |
|---|---|---|---|---|
| BIS-11A | | | | |
| Total score | 57.19 ± 8.596 | 61.39 ± 9.429 | −2.588 | 0.011 |
| Motor impulsiveness | 21.50 (20.00-24.25) | 22.50 (21.00-25.25) | −1.417 | 0.156 |
| Attention impulsiveness | 12.00 (10.00-14.00) | 13.00 (11.00-15.25) | −2.106 | 0.035 |
| Non-planning impulsiveness | 22.69 ± 4.779 | 24.97 ± 4.949 | −2.603 | 0.010 |

Note: BIS-11A, Barratt Impulsiveness Scale; AT, systematic antidepressant treatment; NT, no systematic antidepressant treatment

**Table 4. Correlation analyses between key study variables.**

| | 1 | 2 | 3 | 4 | 5 | 6 | 7 |
|---|---|---|---|---|---|---|---|
| 1. Antidepressant use | 1 | | | | | | |
| 2. Positive affect | −0.037 | 1 | | | | | |
| 3. Negative affect | 0.216* | 0.182* | 1 | | | | |
| 4. BIS-11A total score | 0.245** | −0.114 | −0.102 | 1 | | | |
| 5. Motor impulsiveness | 0.128 | −0.144 | −0.260** | 0.729** | 1 | | |
| 6. Attention impulsiveness | 0.190* | −0.047 | −0.034 | 0.689** | 0.307** | 1 | |
| 7. Non-planning impulsiveness | 0.228* | −0.118 | −0.031 | 0.856** | 0.409** | 0.447** | 1 |

Note: * p < 0.05 (two-tailed test); ** p < 0.01 (two-tailed test); PANAS, Positive and Negative Affect Scale; BIS-11A, Barratt Impulsiveness Scale.

**Table 5. Mediation analyses of positive/negative affect.**

| Mediators | Dependent variables | Total effect, *ES (95%CI)* | Indirect effect, *ES (95%CI)* | Direct effect, *ES (95%CI)* |
|---|---|---|---|---|
| Positive affect | Full impulsiveness | 4.357** (1.164, 7.551) | −0.140 (−0.806, 0.450) | 4.497** (1.340, 7.655) |
| | Attention impulsiveness | 1.234* (0.212, 2.255) | −0.020 (−0.188, 0.087) | 1.254* (0.230, 2.278) |
| | Motor impulsiveness | 0.807 (−0.579, 2.193) | −0.069 (−0.370, 0.218) | 0.876 (−0.488, 2.240) |
| | Non-planning impulsiveness | 2.316** (0.592, 4.041) | −0.051 (−0.318, 0.180) | 2.367** (0.647, 4.088) |
| Negative affect | Full impulsiveness | 4.357** (1.164, 7.551) | −0.459 (−1.336, 0.205) | 4.817** (1.572, 8.061) |
| | Attention impulsiveness | 1.234* (0.212, 2.255) | 0.044 (−0.194, 0.349) | 1.190* (0.144, 2.236) |
| | Motor impulsiveness | 0.807 (−0.579, 2.193) | −0.398 (−0.983, −0.030) | 1.205 (−0.168, 2.577) |
| | Non-planning impulsiveness | 2.316** (0.592, 4.041) | −0.106 (−0.493, 0.331) | 2.422** (0.658, 4.187) |

Note: * p < 0.05 (two-tailed test); ** p < 0.01 (two-tailed test); ES, effect size; 95%CI, 95% confidence interval.

## Discussion

The current study aimed to investigate the influence of systematic antidepressant treatment in the early stages on positive/negative affect and impulsivity in patients with bipolar euthymic disorder. It also sought to explore the potential mediating effects of positive/negative affect in the relationship between systematic antidepressant treatment and the strength of impulsivity. The results show that patients in the AT group exhibited significantly higher levels of negative affect, attentional impulsivity, non-planning impulsivity, and overall impulsivity compared to the NT group during the euthymic period. Additionally, an inverse and significant correlation was found between negative affect and motor impulsivity. However, the affective state (negative affect) did not play a mediating role in the relationship between antidepressant use and dysregulated impulsivity.

Dysregulated impulsivity is a salient feature of BD associated with various negative consequence, such as suicidal attempts, emphasizing the importance of comprehensively understanding the factors that influence its perpetration. Much attention has been given to this domain, and a large body of work has been done. One study by Titone and colleagues showed that BD patients exhibited stronger impulsivity than healthy controls in terms of behavioral, self-report, and daily impulsivity. Further analysis indicated a reciprocal relationship between high impulsivity and high next-day negative affect, but not with positive affect [7]. Similarly, increased negative emotional states have been linked to greater impulsivity in Alzheimer#39;s disease [19], but not in individuals with frontotemporal dementia, although the latter were more impulsive [19]. The results indicated that there may be other factors playing important role in regulating impulsivity, which are involved in the pathway from the negative effect to impulsivity. Additionally, research on emotion dysregulation showed that lifetime suicidal ideation was significantly and positively associated with negative emotion and mood instability [20]. Based on the aforementioned information, it seems that there is a deterministic connection between negative affect and increased impulsivity or suicidal behaviors. Contrastingly, no direct link was observed between negative affect and impulsivity among alcohol abusers, though both are related to suicidal attempts [21]. In our study, negative affect was inversely and significantly associated with motor impulsivity, while positive affect was not correlated with any form of impulsivity. This finding diverges from expectations based on prior literature suggesting that negative emotions instigate impulsive behaviors [7,11]. Potential explanations for this discrepancy could include differences in clinical samples, specific forms of impulsivity assessed, euthymic state, or the unique context of antidepressant treatment at an early stage. For instance, Titone's study compared BD patients to healthy control, with BD patients in remission or experiencing manic or depressive episodes [7], while the participants in our study were all in euthymic state. Low levels of negative affect during the bipolar euthymic phase may be insufficient to trigger a noticeable manifestation of motor impulsivity. The previous study found a significant difference in motor impulsivity between euthymic

bipolar and unipolar disorder patients, but not in attention or non-planning impulsivity [4]. This suggests that the affective state of patients is an important factor when interpreting and comparing impulsivity in individuals with affective disorder, even in remission. Moreover, the discrepancy in impulsivity and affective state between the AT and NT groups highlighted the impact of systematic antidepressant treatment in the early stages. Antidepressants have potential to alter mood states, such as triggering a manic switching from bipolar depression or inducing subthreshold mood fluctuation in unipolar depression [14,15], which may also contribute to the divergence. There are still many factors that influence affective state and impulsivity, such as personality traits [22], cognitive impairment [23], sleep quality [24], which have not been fully evaluated in studies and could account for discrepancies in the results. Finally, our failed mediation analysis indicated that the affective state did not play a regulatory role in the effect of early-stage antidepressant use on dysregulated impulsivity, although significant associations were found between the affective state, systematic antidepressant treatments, and impulsivity. Euthymia was defined as both MADRS and YMRS scores being less than 7, and the participants in the current study exhibited less severe affective symptoms. This may explain the lack of mediating role of the affective state in this context.

A key strength of the current study is that it established a bridge between the early use of antidepressants and their subsequent negative influence on affective state and impulsivity, even several years later. However, the cross-sectional design limits causal inferences. Longitudinal studies are needed to further delineate the temporal and dynamic interplay between antidepressants, affective states, and impulsive tendencies across bipolar mood phases. Another limitation was the focus on self-report measures of affect and impulsivity, which could have been complemented by behavioral or neuropsychological tasks. Additionally, the exploration of potential moderating variables, such as medication adherence, comorbidities, and illness characteristics, would enable a more nuanced understanding. Finally, there are differences between SSRIs and SNRIs in the risk of inducing manic switches in bipolar depression. However, the current study had not conducted further analyses on the effects of different types of antidepressants on positive/negative affect and impulsivity. A larger sample size, with subgroup analyses based on antidepressant class used in early stages, would facilitate a better understanding of the potential long-term effects.

This study establishes a connection between the early use of antidepressants and their subsequent negative influence on affective state and impulsivity, and highlights the importance of careful consideration when prescribing antidepressants to patients with potential to develop BD, as well as monitoring affective states and impulsivity even during euthymic periods. Furthermore, patients at risk of BD may benefit from use of antidepressants with a low risk of inducing manic switching in the early stages. This also emphasizes the need for early diagnosis of BD and personalized treatment plans that account for the potential risks and benefits of antidepressant use in this population. Longitudinal, multimodal research is needed to clarify the directionality and more nuanced relationships between antidepressants, affective experiences, and impulsive behaviors across bipolar affective states.

## Supporting information

**S1 Data. Sociodemographic and clinical characteristics.**
(XLSX)

**S2 Data. Unadjusted mediation analysis.**
(XLSX)

**S3 Data. Adjusted mediation analysis.**
(XLSX)

## Acknowledgments

We appreciate contributions of all participants to this study.

## Author contributions

**Conceptualization:** Chenghao Yang, Jie Li.

**Data curation:** Dongyu Han, Peng Cui.

**Formal analysis:** Mingjin Wang, Xiaofei Hou.

**Funding acquisition:** Chenghao Yang, Jie Li.

**Methodology:** Mingjin Wang, Xiaofei Hou.

**Writing – original draft:** Dongyu Han, Peng Cui.

**Writing – review & editing:** Chenghao Yang, Jie Li.

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
