## [Decision Letter · Decision Letter 0]

19 Sep 2025

Dear Dr. Yang,

Thank you for submitting your manuscript to PLOS ONE. After careful consideration, we feel that it has merit but does not fully meet PLOS ONE’s publication criteria as it currently stands. Therefore, we invite you to submit a revised version of the manuscript that addresses the points raised during the review process.

**ACADEMIC EDITOR: Please insert comments here and delete this placeholder text when finished.**   I kindly ask you to carefully address all the reviewers’ comments and suggestions, ensuring that your revisions are precise, well-argued, and fully integrated into the manuscript. Additionally, I encourage you to elaborate more explicitly on the potential practical implications and future applications of your findings. A thoughtful discussion in this regard would greatly enhance the overall relevance and translational value of your work.

We look forward to receiving your revised manuscript.

Kind regards,

Giuseppe Marano

Academic Editor

PLOS ONE

Journal Requirements:

“The study was financially supported by funds from “Research Plan Project of Tianjin Municipal Education Commission” (2022KJ264), “Tianjin Key Medical Discipline (Specialty) Construction Project” (TJYXZDXK-033A), and “Beijing-Tianjin-Hebei Basic Research Cooperation Project” (23JCZXJC00230, J230011)”

Reviewers' comments:

Reviewer's Responses to Questions

**Comments to the Author**

1. Is the manuscript technically sound, and do the data support the conclusions?

Reviewer #1: Yes

Reviewer #2: Yes

2. Has the statistical analysis been performed appropriately and rigorously?

Reviewer #1: Yes

Reviewer #2: Yes

3. Have the authors made all data underlying the findings in their manuscript fully available?

Reviewer #1: Yes

Reviewer #2: Yes

4. Is the manuscript presented in an intelligible fashion and written in standard English?

Reviewer #1: Yes

Reviewer #2: Yes

Reviewer #1: Thank you for allowing the opportunity to review this manuscript.

Overall - Well-designed and clearly written cross-sectional study addressing a clinically important and relevant question: the long-term impact of early antidepressant treatment on affect and impulsivity in euthymic patients with Bipolar Disorder (BD). Good patient selection criteria and robust statistical analysis.

The finding that early antidepressant use is associated with higher negative affect and impulsivity, even years later during a period of euthymia, is quite significant!

While the study is well-conducted, some key areas require clarification to strengthen the manuscript's conclusions and impact. The primary recommendation is for a Major Revision with the points outlined below.

1. Insufficient Definition of the Independent Variable: The manuscript's primary weakness is the vague definition of its main independent variable: "systematic antidepressant treatment." Authors defined it as "receiving an adequate dosage of antidepressants for at least 6 weeks". I believe this definition is insufficient for proper interpretation and replication.

Suggestion: The authors must provide a more detailed description which should include: specific antidepressants classes (SSRIs, SNRIs, TCAs), dose range considered "adequate", average treatment duration in the AT group, was it monotherapy or adjunctive treatments. This information is crucial, as different antidepressants have varying risks for inducing mood switch and may have different long-term effects.

2. Justification for DSM-IV-TR: Author’s have used the DSM-IV-TR for diagnosis, even though study began in 2020 (DSM-5 was published in 2013). While potentially valid, the choice of an older diagnostic manual requires justification.

Suggestion: Explain the rationale for using DSM-IV-TR in the Methods section (? to maintain consistency with historical data / EMR related? from the institution, standardized use of the SCID-IV at the time of study inception, etc.).

3. Interpretation of Unexpected Findings: In the results, authors show a significant inverse correlation between negative affect and motor impulsivity (r=−0.260, p=0.004). This is a counterintuitive finding, as higher negative affect is often presumed to increase impulsivity. The Discussion section does not adequately explain its potential meaning.

Suggestion: Expand the discussion to speculate on possible reasons for this inverse relationship. Could higher levels of residual negative affect in euthymic patients manifest as anxiety or rumination that inhibits motor action, while cognitive forms of impulsivity (attentional, non-planning) remain elevated?

Minor Suggestions

Terminology Consistency: The terms "systematic" and "systemic" have been used interchangeably throughout the manuscript. Please replace all instances of "systemic" with "systematic" for consistency.

Figure 2 displaying the mediation analysis model, is very dense and difficult to interpret quickly. It presents multiple models simultaneously, which may confuse readers. Consider replacing it with a table that clearly presents the results of the mediation analyses (i.e., direct, indirect, and total effects with confidence intervals for each impulsivity subtype). A simplified conceptual diagram could be retained if desired.

Typographical Errors:

Abstract, Conclusions: "There findings highlight..." should be "These findings highlight...".

Discussion, Paragraph 1: "...exhibited significantly ihigher levels..." should be "...exhibited significantly higher levels...".

By addressing these points, the authors can significantly improve the clarity, impact, and scientific rigor of their manuscript.

Best regards.

Reviewer #2: A fairly comprehensive job done. Can extend it further by conducting a longitudinal study to ascertain whether the impulsivity and negative affectivity changes in subsequent phases of euthymia. Also, whether there had been a change in pattern of substance use in these euthymic periods may have been determined.

.

Reviewer #1: **Yes:**Vishesh AgarwalVishesh AgarwalVishesh AgarwalVishesh Agarwal

Reviewer #2: **Yes:**Rudraprasad AcharyaRudraprasad AcharyaRudraprasad AcharyaRudraprasad Acharya

---

## [Author Response · Author response to Decision Letter 1]

27 Sep 2025

Point by point response to reviewers

Reviewer 1.

Overall - Well-designed and clearly written cross-sectional study addressing a clinically important and relevant question: the long-term impact of early antidepressant treatment on affect and impulsivity in euthymic patients with bipolar disorder (BD). Good patient selection criteria and robust statistical analysis. The finding that early antidepressant use is associated with higher negative affect and impulsivity, even years later during a period of euthymia, is quite significant!

1. Insufficient Definition of the Independent Variable: The manuscript's primary weakness is the vague definition of its main independent variable: "systematic antidepressant treatment." Authors defined it as "receiving an adequate dosage of antidepressants for at least 6 weeks". I believe this definition is insufficient for proper interpretation and replication.

Suggestion: The authors must provide a more detailed description which should include: specific antidepressants classes (SSRIs, SNRIs, TCAs), dose range considered "adequate", average treatment duration in the AT group, was it monotherapy or adjunctive treatments. This information is crucial, as different antidepressants have varying risks for inducing mood switch and may have different long-term effects.

Response:

Thanks a lot for reviewer’s constructive suggestion regarding the definition. We have revised it as the reviewer suggested. Based on the study procedure, we recruited patients treated with SSRIs or SNRIs as monotherapy, including sertraline 50-200mg/d, fluoxetine 20-60mg/d, paroxetine 20-60mg/d, citalopram 20-60mg/d, escitalopram 10-20mg/d, venlafaxine 75-225mg/d, duloxetine 30-120mg/d. Regarding the dosage range, we considered that doses with antidepressant effects may carry the risk of inducing mania. So, we selected the minimum effective dose to the maximum dose allowed by the product label. It is possible that this information differs between Mainland China and other countries. No restrictions were placed on benzodiazepines, zolpidem, zopiclone, zaleplon, used as adjuncts to improve sleep. Additionally, given the difference in risk to induce mood switch, we have added content about it in limitations.

Actions: (line 111-117, page 6,7; line 309-314, page 19)

1. …the definition of “systematic antidepressant treatment” is that patients receive treatment with SSRIs or SNRIs as monotherapy, including sertraline 50-200mg/d, fluoxetine 20-60mg/d, paroxetine 20-60mg/d, citalopram 20-60mg/d, escitalopram 10-20mg/d, venlafaxine 75-225mg/d, duloxetine 30-120mg/d, for at least 6 weeks. No restrictions were placed on benzodiazepines, zolpidem, zopiclone, zaleplon, used as adjuncts to improve sleep...

2. Finally, there are differences between SSRIs and SNRIs in the risk of inducing manic switches in bipolar depression. However, the current study had not conducted further analyses on the effects of different types of antidepressants on positive/negative affect and impulsivity. A larger sample size, with subgroup analyses based on antidepressant class used in early stages, would facilitate a better understanding of the potential long-term effects.

2. Justification for DSM-IV-TR: Author’s have used the DSM-IV-TR for diagnosis, even though study began in 2020 (DSM-5 was published in 2013). While potentially valid, the choice of an older diagnostic manual requires justification.

Suggestion: Explain the rationale for using DSM-IV-TR in the Methods section (? to maintain consistency with historical data / EMR related? from the institution, standardized use of the SCID-IV at the time of study inception, etc.).

Response:

Thanks for reviewer’s valuable suggestion. As the reviewer mentioned, the DSM-5 has been widely adopted, and the use of standardized diagnostic criteria facilitates the interpretation and generalization of research findings. But the current study was conceived in 2019 and conducted in mid-2020, at which point the DSM-5 and SCID-5 had not yet been implemented in China. Additionally, the diagnostic criteria for bipolar disorder remained largely unchanged between the DSM-IV-TR and DSM-5, with the exception of the reclassification of the mixed episode as mixed features. Therefore, we would say that the use of DSM-IV-TR in this study did not introduce significant bias in the recruitment of bipolar patients. We have provided the corresponding explanation in the Methods section, as the reviewer suggested.

Actions: (line 123-124, page 7)

…the study was conceived in 2019 and launched in mid-2020, at which point the DSM-5 and SCID-5 had not yet been implemented in China…

3. Interpretation of Unexpected Findings: In the results, authors show a significant inverse correlation between negative affect and motor impulsivity (r=−0.260, p=0.004). This is a counterintuitive finding, as higher negative affect is often presumed to increase impulsivity. The Discussion section does not adequately explain its potential meaning.

Suggestion: Expand the discussion to speculate on possible reasons for this inverse relationship. Could higher levels of residual negative affect in euthymic patients manifest as anxiety or rumination that inhibits motor action, while cognitive forms of impulsivity (attentional, non-planning) remain elevated?

Response:

Thank the reviewer for raising this important issue. As illustrated in the discussion, many studies looked at the relationship between negative affect and impulsivity and showed that diverse factors are involved in the regulation of negative affect. But no study was conducted in the context of euthymic bipolar disorder, so we attempted to reason this conclusion within an appropriate scope (line 262-283). Bipolar disorder symptoms are complex, for example, mixed features can involve both manic and depressive symptoms simultaneously. And there is a significant difference in motor impulsivity between euthymic bipolar and unipolar disorder patients. Therefore, we have reason to believe that even during remission, subthreshold symptoms may complicate the situation. Additionally, although there is no significant correlation between positive emotions and motor impulsivity, it cannot be ruled out that, in the absence of sufficiently strong negative emotions during remission, positive emotions might counterbalance the eventual manifestation of impulsivity derived from negative affect. As you suggested, anxiety or rumination derived from relative higher levels of residual negative affect in euthymic patients may inhibit motor action. But we found it is difficult to systematically elaborate on this result based on speculation, like sometimes anxiety can instigate the motor activity. In this regard, we tried to add one sentence. And could you please reconsider the rationality of the current explanation, specifically on the content from line 274-283. Thank you so much.

Actions: (line 282-283, page 18)

Low levels of negative affect during the bipolar euthymic phase may be insufficient to trigger a noticeable manifestation of motor impulsivity.

4. Minor Suggestions: Terminology Consistency: The terms "systematic" and "systemic" have been used interchangeably throughout the manuscript. Please replace all instances of "systemic" with "systematic" for consistency.

Response:

Thanks for reviewer’s careful reading. We have adjusted it as you suggested for ten instances totally.

5. Minor Suggestions: Figure 2 displaying the mediation analysis model, is very dense and difficult to interpret quickly. It presents multiple models simultaneously, which may confuse readers. Consider replacing it with a table that clearly presents the results of the mediation analyses (i.e., direct, indirect, and total effects with confidence intervals for each impulsivity subtype). A simplified conceptual diagram could be retained if desired.

Response:

Thanks a lot for reviewer’s suggestions. We have drawn a table (Table 5) instead of Fig 2, to present the details of mediation analyses (line 245-246, page 16).

6. Typographical Errors: Abstract, Conclusions: "There findings highlight..." should be "These findings highlight...". Discussion, Paragraph 1: "...exhibited significantly ihigher levels..." should be "...exhibited significantly higher levels...".

Response:

Thank you again for your thoughtful instruction, and we apologize for our oversight. We have revised them.

Reviewer 2.

A fairly comprehensive job done. Can extend it further by conducting a longitudinal study to ascertain whether the impulsivity and negative affectivity changes in subsequent phases of euthymia. Also, whether there had been a change in pattern of substance use in these euthymic periods may have been determined.

Response:

Thank the reviewer for this insightful suggestion. The lack of early diagnostic methods for bipolar disorder has led to the widespread use of antidepressants in the early stages, and it is important to investigate what long-term impacts such treatment might have on patients with bipolar disorder. Based on the current cross-sectional study, we have preliminarily observed the long-term effects of early antidepressant pharmacotherapy on the positive and negative emotions as well as impulsivity in individuals with bipolar disorder. As the reviewer mentioned, longitudinal studies would be more beneficial for better assessing the reliability and validity of the current findings. This is also a key focus of our future research.

---

## [Decision Letter · Decision Letter 1]

26 Mar 2026

A Cross-Sectional Study on the Impact of Early Systematic Antidepressant Therapy on Positive/Negative Affect and Impulsivity in Euthymic Bipolar Disorder Patients

PONE-D-25-10699R1

Dear Dr. Yang,

We’re pleased to inform you that your manuscript has been judged scientifically suitable for publication and will be formally accepted for publication once it meets all outstanding technical requirements.

Kind regards,

Giuseppe Marano

Academic Editor

PLOS One

Additional Editor Comments (optional):

Reviewers' comments:

Reviewer's Responses to Questions

**Comments to the Author**

Reviewer #1: All comments have been addressed

Reviewer #3: All comments have been addressed

Reviewer #4: All comments have been addressed

2. Is the manuscript technically sound, and do the data support the conclusions?

Reviewer #1: Yes

Reviewer #3: Yes

Reviewer #4: Yes

3. Has the statistical analysis been performed appropriately and rigorously?

Reviewer #1: Yes

Reviewer #3: Yes

Reviewer #4: Yes

4. Have the authors made all data underlying the findings in their manuscript fully available?

Reviewer #1: Yes

Reviewer #3: Yes

Reviewer #4: Yes

5. Is the manuscript presented in an intelligible fashion and written in standard English?

Reviewer #1: Yes

Reviewer #3: Yes

Reviewer #4: Yes

Reviewer #1: Dear Authors,

Thank you for your thorough and thoughtful revision of the manuscript. I'm very pleased with the changes you've made, and believe you have successfully addressed all of the points raised in the previous review. The revisions have substantially improved its quality, and the manuscript is now much stronger, clearer, and makes an even more compelling contribution to the field. I am pleased to recommend it for publication.

Reviewer #3: This is an excellent article that explores the treatment of bipolar disorder and holds significant value for the therapy of patients with bipolar disorder.

Reviewer #4: In the first round of review, the issues with the manuscript were mainly focused on the definition of concepts, outdated diagnostic criteria, and the accuracy of wording. The authors provided detailed explanations and made corresponding revisions, resulting in clearer expression and more coherent content. Regarding comment 3, the authors’ explanation is not perfect but acceptable, representing a reasonable interpretation of study results. Systematic treatment with antidepressants in the early stages of bipolar disorder is indeed common, yet relevant studies are scarce and there is insufficient data for comparative reference, which to some extent limits the interpretation of this issue. However, the current study can serve to highlight that this field deserves more attention.

.

Reviewer #1: **Yes:**Vishesh AgarwalVishesh AgarwalVishesh AgarwalVishesh Agarwal

Reviewer #3: **Yes:**Ruijun HanRuijun HanRuijun HanRuijun Han

Reviewer #4: No

---

## [Editor Report · Acceptance letter]

PONE-D-25-10699R1

PLOS One

Dear Dr. Yang,

I'm pleased to inform you that your manuscript has been deemed suitable for publication in PLOS One. Congratulations! Your manuscript is now being handed over to our production team.

Kind regards,

on behalf of

Dr. Giuseppe Marano

Academic Editor

PLOS One